

# The Effect of Amino Acids on the Fenton and photo-Fenton Reactions in Cloud Water: Unraveling the Dual Role of Glutamic Acid

*Peng Cheng [1,2], Gilles Mailhot [1,3], Mohamed Sarakha [1], Guillaume. Voyard [1], Daniele Scheres Firak [4], Thomas Schaefer [4], Hartmut Herrmann [4], Marcello Brigante [1]\**

[1] Université Clermont Auvergne, CNRS, Institut de Chimie de Clermont-Ferrand, F-63000 Clermont–Ferrand, France.

[2] Department of Environmental Engineering, School of Resources and Environmental Science, Wuhan University, 430079, PR China

[3] Université Clermont Auvergne, CNRS, Laboratoire de Météorologie Physique (LaMP), F-63000 Clermont–Ferrand, France.

[4] Atmospheric Chemistry Department (ACD), Leibniz- Institute for Tropospheric Research (TROPOS), 04318 Leipzig, Germany

*Corresponding author:

Marcello Brigante (Marcello.Brigante@uca.fr)



**ABSTRACT**
In this work, Glutamic acid (Glu) was selected as a model amino acid (AAs) to investigate its
complexation with Fe(III) and Fe(II), focusing on its impact on the Fenton reaction and the
photolysis of Fe(III) in cloud aqueous phase. Glu was found to enhance the rate constant for the
reaction of Fe(II)-Glu with $H_2O_2$ to $1.54 \pm 0.13 \times 10^4$ $M^{-1}$ $s^{-1}$, which is significantly higher than
that of classic Fenton reactions (~50-70 $M^{-1}$ $s^{-1}$). In contrast, the photolysis quantum yield of
Fe(III)-Glu complex was determined to be 0.037 under solar simulated irradiation, largely lower
than Fe(III)-hydroxy complexes (0.216). In the overall process (Fenton or Fe(III) photolysis),
it was found that $^{\bullet}OH$ formation decreased in the presence of Glu. Additionally, the fate of Glu
in the presence of Fe(III) was investigated as well as the oxidation process (driven by $^{\bullet}OH$ and
ligand-to-metal charge transfer (LMCT) reaction) led to the formation of short-chain carboxylic
acids and ammonium under simulated solar light. Interestingly, these two processes generated
different primary short-chain carboxylic acids, indicating distinct mechanisms. This study
provides valuable insights into the role and fate of amino acids in atmospheric chemistry,
helping to further understand their impact on atmospheric processes.

**KEYWORDS**: Glutamic acid, Fenton, hydroxyl radical, oxidant capacity, atmospheric
composition

**SYNOPSIS**
This study investigates the complexation of Fe(II) and Fe(III) with glutamic acid under cloud
water conditions and the effect on Fenton and photo-Fenton reactions, hydroxyl radical
formation, and their impact on amino acid oxidation.





## 1. INTRODUCTION


The Earth's atmosphere is a dynamic system in which different phases, including gases, aerosol
particles, water droplets, and ice particles, are all engaged in complex chemical interactions that
continually modify the atmospheric chemical composition (Bianco et al., 2020; Kanakidou et
al., 2018). Among these, the cloud aqueous phase stands out as a critical reactive system,
encompassing gaseous, liquid, and solid components. In recent years, intensified research
efforts have centered on unraveling the composition of atmospheric cloud waters, significantly
advancing our comprehension of multiphase chemistry within the atmosphere (Bianco et al.,
2018). Common components identified in both aerosols and cloud water include inorganic ions,
transition metal ions (TMI) (Angle et al., 2021; Bianco et al., 2017), and organic carbon
(Battaglia Jr. et al., 2019).
Recent investigations have unveiled the presence of reactive oxygen species (ROS) in viscous
aerosol particles, highlighting their pronounced reactivity in such environments (Alpert et al.,
2021; Edwards et al., 2022). Hydroxy radicals ($^{\bullet}$OH) emerge as primary ROS in the atmospheric
water phase, with concentrations estimated between $10^{-14}$ to $10^{-12}$ M$^{-1}$ (Bianco et al., 2020;
Gligorovski et al., 2015). Key sources of $^{\bullet}$OH include gas-droplet partitioning and in situ
formation through processes like photolysis at surfaces or in the bulk phase, such as the
photolysis of TMI and hydrogen peroxide ($H_2O_2$) (Bianco et al., 2015; Tilgner et al., 2013).
Iron (Fe), copper (Cu), and manganese (Mn) have gained prominence as pivotal metals in
atmospheric chemical processes due to their elevated concentrations, with Fe averaging around
$10^{-6}$ M in the atmospheric aqueous phase (Sorooshian et al., 2013). Experimental evidence and
literature emphasize the crucial role of iron, particularly via (photo)-Fenton and (photo)-Fenton-



like processes, in the generation and budgeting of $^\bullet$OH (R1-R5) (Guo et al., 2014; Tilgner et al.,

65    2013).

$$[Fe(H_2O)_6]^{3+} \xrightarrow{h\nu} [Fe(H_2O)_5]^{2+} + H^+ + {}^\bullet OH \qquad (R1)$$
$$[Fe(H_2O)_5OH]^{2+} \xrightarrow{h\nu} [Fe(H_2O)_5]^{2+} + {}^\bullet OH \qquad (R2)$$
$$[Fe(H_2O)_5]^{2+} + H_2O_2 \rightarrow [Fe(H_2O)_5OH]^{2+} + {}^\bullet OH \qquad (R3)$$
$$[Fe(H_2O)_6]^{3+} + H_2O_2 \rightarrow [Fe(H_2O)_5]^{2+} + HO_2{}^\bullet + H_2O + H^+ \qquad (R4)$$
$$[Fe(H_2O)_5OH]^{2+} + H_2O_2 \rightarrow [Fe(H_2O)_5]^{2+} + HO_2{}^\bullet + H_2O \qquad (R5)$$
While Fe(III)/Fe(II) ions precipitate as oxides or hydroxides at pH higher than 4.0, in the cloud
water phase, iron complexes with organic ligands enhance stability under typical cloud water
photooxidation conditions (Soriano-Molina et al., 2018; Yuan et al., 2020). Various organic
ligands, including carboxylic acids and aldehydes, have been extensively studied (Long et al.,
2013; Marion et al., 2018; Soriano-Molina et al., 2018). However, less than 30 % of the
dissolved organic carbon (DOC) in the cloud-aqueous phase has been molecularly characterized,
with amino acids (AAs) constituting a significant portion of DOC (Bianco et al., 2016).
Numerous field studies have confirmed the presence of AA in cloud water, rain, fog, and
aerosols, with concentrations typically ranging from low nanomolar to micromolar levels,
depending on the location and sampling method (Matos et al., 2016; van Pinxteren et al., 2023;
Renard et al., 2022; Triesch et al., 2021). For example, Renard et al. (2022) detected more than
15 amino acids in cloud water collected at Puy de Dôme, France, with glutamate being one of
the most abundant species. These compounds originate from both primary emissions (e.g.,
bioaerosols, ocean spray) and secondary atmospheric processes (e.g., processing of proteins or
peptides within clouds) (Mace et al., 2003; Samy et al., 2011). Amino acids, as key nitrogen-
containing components in organic matter, can significantly affect the oxidation capacity of





cloud water through free radical scavenging and metal complexation reactions (Bianco et al.,
2016; Marion et al., 2018), but their specific atmospheric reactivity and transformation
mechanisms are still unclear. The photochemical behavior and fate of AAs in the atmosphere
remain relatively unexplored. For example, tryptophan can undergo direct photolysis,
producing low-molecular-weight compounds and dimerization products under solar-simulated
conditions. Recent investigations into the fate of the Fe(III)-aspartate complex demonstrate
ligand-to-metal charge transfer reactions (LMCT) and the formation of ammonia and short-
chain carboxylic acids (Marion et al., 2018).
However, the effect of the complexation between Fe(II) and AAs on the rate of Fenton reaction
and the yield of $^\bullet$OH in the atmosphere has not yet been investigated. Moreover, the effect of
the complexation between Fe(III) and AAs on the quantum yield of atmospheric photolysis of
Fe(III) deserves further investigation, since both processes highly affect the budget of $^\bullet$OH
during the day and night in the atmosphere. In addition, the complexation between Fe(III) and
AAs introduces two distinct photooxidation pathways: the photolysis of Fe-AAs complexes and
reactions between AAs and (photo)-generated $^\bullet$OH. Although both pathways significantly
contribute to the transformation of AAs in cloud water and impact inorganic and organic
chemical compositions, their mechanisms still lack further study, especially in terms of products
generation.
This study specifically focuses on glutamic acid (Glu), an AA regularly detected in cloud water
and aerosols (van Pinxteren et al., 2012; Triesch et al., 2021), and on the investigation of its
impact on iron (Fe(II)/Fe(III)) reactivity. The study explored i) the effect of Glu on the rate and
$^\bullet$OH yield of the Fenton reaction; and ii) the effect of Glu on the $^\bullet$OH production and Fe(II)
quantum yield during the Fe(III) photolysis. In addition, the study explores iii) two pathways



of Fe(III) and Fe(III)-Glu complex photolysis: the LMCT process and the reaction between Glu
and $^\bullet$OH, assessing their respective contributions to Glu fate. Utilizing competitive kinetic
experiments, the contributions of each pathway were estimated, and a detailed investigation of
the formation, and chemical mechanisms of transformation products was carried out. Ultimately,
our study aims to quantify the diverse contributions of different pathways in amino acid
conversion in the presence of iron.
**2. MATERIAL AND METHODS**
**2.1. Chemicals**
All chemicals were used without further purification: Fe(III)-perchlorate (99.9 %), Fe(II)-
perchlorate (99.9 %), L-glutamic acid monosodium salt (Glu, 99 %), hydrogen peroxide ($H_2O_2$,
30 %), malonic acid (99.0 %), and 2,4-dinitrophenylhydrazine (DNPH, 97 %) were purchase
from Sigma Aldrich. Sodium formate (99.0 %), potassium oxalate monohydrate (99.0 %),
sodium succinate dibasic (98.0 %), and 3-(2-pyridyl)-5,6-diphenyl-1,2,4-triazine-p, p'-sulfonic
acid monosodium salt hydrate (Ferrozine, 97 %) were purchased from Fluka. Ammonium
acetate (99.3 %) was purchased from Fisher. Water was purified using a reverse osmosis RIOS
5 and Synergy (Millipore) device (resistivity 18.2 M$\Omega$ cm, DOC < 0.1 mg L$^{-1}$). All solutions
were prepared in milli-Q water.
**2.2. Experimental procedure**
**2.2.1. Fenton reaction**
The Fenton experiments were carried out with Fe(II) perchlorate at room temperature and a pH
of 5.6 ± 0.1 (Kinetic experiments) and 3.8 ± 0.1 (Electron spin resonance (ESR) experiments).
The Fenton kinetic experiments were initiated by the addition of the $H_2O_2$. The solution was



continuously stirred during the reaction. The pH of the solution was adjusted using $HClO_4$ or
NaOH solutions. The samples were taken every 15 seconds and mixed with a solution of
Ferrozine in phosphate buffer (pH = 7.0 ± 0.1 ) (Gabet et al., 2023). Phenol was used as $^\bullet OH$
scavenger in the experiment. As a scavenger, the required concentration of phenol was
calculated to quench $^\bullet OH$ so that theoretically 99 % of $^\bullet OH$ can be trapped via reacting with
phenol. The same method was used in the presence of Glutamic acid (Glu) to study the Fe(II)-
Glu complex Fenton-like reaction at the same pH. To get different fractions of Fe(II)-Glu, Fe(II)
was mixed with varying concentrations of Glu solution (0 - 25 mM) to calculate the reactivity
constant of the Fenton reaction. The experimental data were analyzed using Origin 2019
software. To determine and quantify the $^\bullet OH$ generation in the Fenton reaction, the ESR
experiment was carried out using 5,5-dimethyl-1-pyrroline-N-oxide (DMPO) as the spin trap.
$Fe(ClO_4)_2$ and $H_2O_2$ were mixed with DMPO at a pH of 3.8 ± 0.1. The pH was set because the
ESR signal intensity was lower at a higher pH = 4.0. ESR spectroscopy was performed on a
Bruker EMX-plus spectrometer using the resonator 4119HS. Detailed information was
provided in the supplementary material section (**SM1**).
**2.2.2. Photolysis of Fe(III)**
To study the Fe(III) photolysis, isopropanol was used as a scavenger in the solution to quench
the generated $^\bullet OH$ radicals. The pH of the solution was adjusted to 3.8 ± 0.1 with $HClO_4$ or
NaOH solutions. The Fe(III) solution was irradiated in a Pyrex jacked cylindrical reactor (**Fig.**
**SM1**) with a circulation cooling system to keep a constant temperature of 283 ± 0.2 K. The
reactor was located at the focal point of a 500 W xenon lamp equipped with a Pyrex filter to
remove wavelengths < 290 nm and a water filter for infrared radiation absorption. The solution
was stirred with a Teflon-coated magnetic stirring bar to ensure homogeneity. The same setup





was used for the photolysis experiments in the presence of Fe(III)-Glu complexes. Different
fractions of Fe(III)-Glu were achieved by adding different amounts of a Glu 50 mM stock
solution (designed [Glu] = 0 - 200 μM).
The emission spectrum of the irradiation setup was recorded using a calibrated CCD camera
(Ocean Optics USB 2000+UV-Vis) coupled with an optical fiber. A total Energy of $8.38 \times 10^3$
$\mu W\ cm^{-2}\ s^{-1}$ was determined between 290 and 500 nm (UV contribution from 290 to 400 nm)
as shown in **Fig. SM2**. The Energy and photonic flux ($I_0$) of the polychromatic irradiation at
every nanometer wavelength are listed in **Table SM1**. Detailed information about the
calculation of the Fe(III) and Fe(III)-Glu photolysis quantum yield is given in the
supplementary material section (**SM2**). To quantify the $^\bullet OH$ generation during the Fe(III)
photolysis, isopropanol was used in excess (10 mM) as a selective $^\bullet OH$ probe. Isopropanol
reacts with $^\bullet OH$ to form acetone which was quantified by HPLC (see section 2.4) (Motohashi
and Saito, 1993).
**2.2.3. Photodegradation of Glu**
To investigate the fate of Glu in various systems, experiments were performed using the
previously described photoreactor setup. Glu solutions, either alone or mixed with Fe(III)
and/or $H_2O_2$, were irradiated under simulated solar light at pH $3.8 \pm 0.1$. Samples were collected
at specific time intervals and analyzed using HPLC-MS (see section 2.4). To calculate and
compare the photodegradation kinetics of Glu in different systems, a pseudo-first-order kinetic
model was applied, expressed as Equation (1):

175          $-\ln(C_t/C_0) = k_{obs}\ t$                                    Eq (1)



where $C_0$ represents the initial concentration of Glu, and $C_t$ is the concentration of Glu at time
$t$ of irradiation. In addition, IC-MS and TOC analyses were performed to identify the generated
by-products and assess the mineralization of Glu (see section 2.4).
**2.3. Study of the speciation of the Fe(III)/Fe(II)-Glu complex**
The speciation of the Fe(III)/Fe(II)-Glu complex was studied using the Hyss 2009 software.
This analysis included the iron, iron-aqua, iron hydroxy, and iron-Glu complexes in the solution.
The parameters used in the software, such as iron and Glu concentrations, kept consistent with
the one in the experimental procedure. The stability constants (log K) used for the complexes,
such as the Fe(II)-Glu and Fe(III)-Glu complexes, etc. are listed in **Table SM2**. These constants
are derived from the Visual MINTEQ database or NIST database 46 and have been corrected
for a temperature of 25 °C and an ionic strength (I) of 0 M. The detailed method is provided in
the supplementary material section (**SM3**).
**2.4. Chemical analysis**
**2.4.1. Fe(II), $H_2O_2$, and Acetone quantification**
Iron (II) concentration was determined by using Ferrozine, which forms a stable magenta
complex with Fe(II) (Fe(II)-ferrozine) (Gabet et al., 2023). Hydrogen peroxide concentration
during experiments was determined by using a spectrofluorimetric quantification method
(Bader et al., 1988). The concentration of generated acetone in the solution was evaluated by
HPLC (Shimadzu NEXERA XR HPL) equipped with a photodiode array detector and an
autosampler (Wang et al., 2005). **Fig. SM3** shows the calibration curve of Fe(II), $H_2O_2$, and
acetone. More details are given in the supplementary material section (**SM4**).
**2.4.2. UPLC-MS, IC-MS and TOC**



The quantification of Glutamic acid (Glu) and the identification of its transformation products
was conducted using a ThermoScientific Orbitrap Q-Exactive high-resolution mass
spectrometry (HRMS) coupled with a ThermoScientific Ultimate 3000 RSLC ultra-high-
performance liquid chromatography (UPLC) system. The quantification of carboxylic acid by-
products and $NH_4^+$ resulting from Glu degradation was performed using a Thermo-Fisher
Scientific ICS-6000 Ionic chromatograph interfaced with a simple quadrupole mass
spectrometer (ISQ-EC-Thermo Scientific). The total organic carbon (TOC) concentration in the
aqueous solution was followed by a Shimadzu TOC 5050A analyzer. Detailed information is
reported in the Supplementary Material section (**SM5**).
**2.5. Kinetic Modeling**
To verify the obtained experimental reactivity constants of the reaction between Fe(II)-Glu and
$H_2O_2$, COPASI software was utilized to simulate the kinetics of Fe(II) consumption and
generation of •OH in the Fenton reaction in the presence of Glu using the default settings of the
deterministic LSODA algorithm to solve ordinary differential equations (Hoops et al., 2006).
The chemical reactions considered in the model are provided in **Table SM3**. The majority of
rate constants used in the model were available in the literature or obtained from experimental
results. For the unknown or uncertain rate constants, the value is obtained from the estimation
according to a similar reaction.
**3. RESULTS AND DISCUSSION**
To investigate the effect of Glu on the Fe(II)/Fe(III) cycle, a complex set of experiments was
performed. First, the complexation of Fe(II)/Fe(III) with Glu was studied as a function of pH
and the initial concentration of Glu. Second, to study the effect of Glu on the Fenton reaction,



its rate constants and $^\bullet$OH generation in the presence of Glu were obtained experimentally and
using the kinetic model. The formation rates of Fe(II) and $^\bullet$OH were determined from Fe(III)
photolysis with or without Glu. Finally, the mechanism of Glu photo-transformation was
reported.
**3.1. Complexation of Glu with Fe(II)/Fe(III)**
The Fe speciation was initially investigated to understand how Glu interacts with iron ions
under various conditions with Hyss2009 software. **Fig. SM4a** shows the speciation of 20 μM
Fe(II) in the presence of Glu (0.2 – 25 mM) across a pH range of 4 to 10. It can be observed
that Fe(II) predominates until pH = 5, while the fraction of the Fe(II)-Glu complex increases
after this pH. Hence, a higher pH (5.6) was selected for the Fenton reaction to guarantee the
presence of complex, while still working under aerosol/cloud conditions and to avoid iron
precipitation occurring at higher pH values. At pH 5.6, the Fe(II)-Glu complex accounts for
2.2 % in the presence of 20 μM Fe(II) and 25 mM Glu. The complex fractions at varying Glu
concentrations at pH = 5.6 are provided in **Table SM4**.
**Fig. SM4b** shows the simulated speciation of Fe(III) (100 μM) as a function of pH in the
presence of Glu (10 -20 μM). The Fe(III)-aqua, Fe(III)-hydroxy complexes, and Fe(III)-Glu
complexes were observed as a function of the pH. At pH = 3.8, [Fe(III)] = [Glu] = 100 μM, the
Fe(III)-hydroxy complexes $Fe(OH)^{2+}$ and $Fe(OH)_2^+$ represent 24.4 and 22.8 % of the total Fe(III)
concentration, respectively. In contrast, Fe(III)-Glu complex accounts for 52.3 % of the total
Fe(III), while Fe(III)-aqua complex constitutes only 0.5 %. The UV-Vis spectra of Fe(III), Glu,
and Fe(III)-Glu complex are depicted in **Fig. SM2**. The characteristic absorption band of Fe(III)
with a maximum at 297 nm, corresponding to the charge transfer bands of $Fe(OH)^{2+}$, becomes
attenuated in the presence of Glu. Moreover, the UV-Vis spectrum of Fe(III)-Glu mixture



differs from those of Fe(III) and Glu alone or the simple overlap of their individual spectra,
confirming the formation of a stable Fe(III)-Glu complex (Samavat et al., 2007). The fractions
of the generated complex in the presence of different Glu concentrations at pH = 3.8 are given
in **Table SM5**. For the sake of simplicity, Fe(III)/Fe(II)-hydroxy and Fe(III)/Fe(II)-aqua
complexes are hereafter referred as Fe(III) and Fe(II).
**3.2. Fenton reaction process in the presence of Glu**
**3.2.1. Fe(II) oxidation**
To study the effect of Glu on the kinetics of the Fenton reaction and determine the rate constant
of the reaction of Fe(II)-Glu with $H_2O_2$, experiments were performed using different
concentrations of Glu. **Fig. 1a** shows the faster Fe(II) concentration decreases when the Glu
concentration increases, which indicates that Glu can increase the reaction rate of Fe(II) with
$H_2O_2$. This is likely due to the formation of the Fe(II)-Glu complex which has a high reaction
rate constant with $H_2O_2$. As seen in **Fig. 1b**, the data obtained by plotting $\frac{-\frac{d[Fe(II)]}{dt}}{[H_2O_2][Fe(II)]}$ as a
function of the fraction of Fe(II)-Glu can be fitted with a linear equation y = ax + b, where a is
equal to $1.54 \pm 0.13 \times 10^4$ $M^{-1}$ $s^{-1}$ and represents the rate constant of reaction of Fe(II)-Glu with
$H_2O_2$, and b is equal to rate constant of Fe(II) with $H_2O_2$ ($-\frac{d[Fe(II)]}{dt}$ data is provided in **Table.**
**SM4**). This value is much higher than the rate constant of the classic Fenton reaction which has
a rate constant of about 50-70 $M^{-1}$ $s^{-1}$ (Kremer, 2003; Neyens and Baeyens, 2003;
Rachmilovich-Calis et al., 2009). The reason behind this increase is likely due to the Fe(II)-Glu
complex accessing a lower reduction potential calculated to be + 0.241 V compared with the
Fe(II) (+0.771 V) (Strathmann and Stone, 2002), which contributes to the higher rate constant
of the reaction of Fe(II)-Glu with $H_2O_2$.

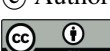

Then the Fenton reaction model was used to fit the experimental data to verify the rate constant
value of the reaction between Fe(II)-Glu and $H_2O_2$. As shown in **Fig. 1a**, the experimental data
of Fe(II) kinetics can be well-fitted by the model. The fitted rate constant value of the reaction
between Fe(II)-Glu and $H_2O_2$ was obtained at a range of $1.2\times10^4$ to $1.8\times10^4$ $M^{-1}$ $s^{-1}$, which is
very close to the experimental results.

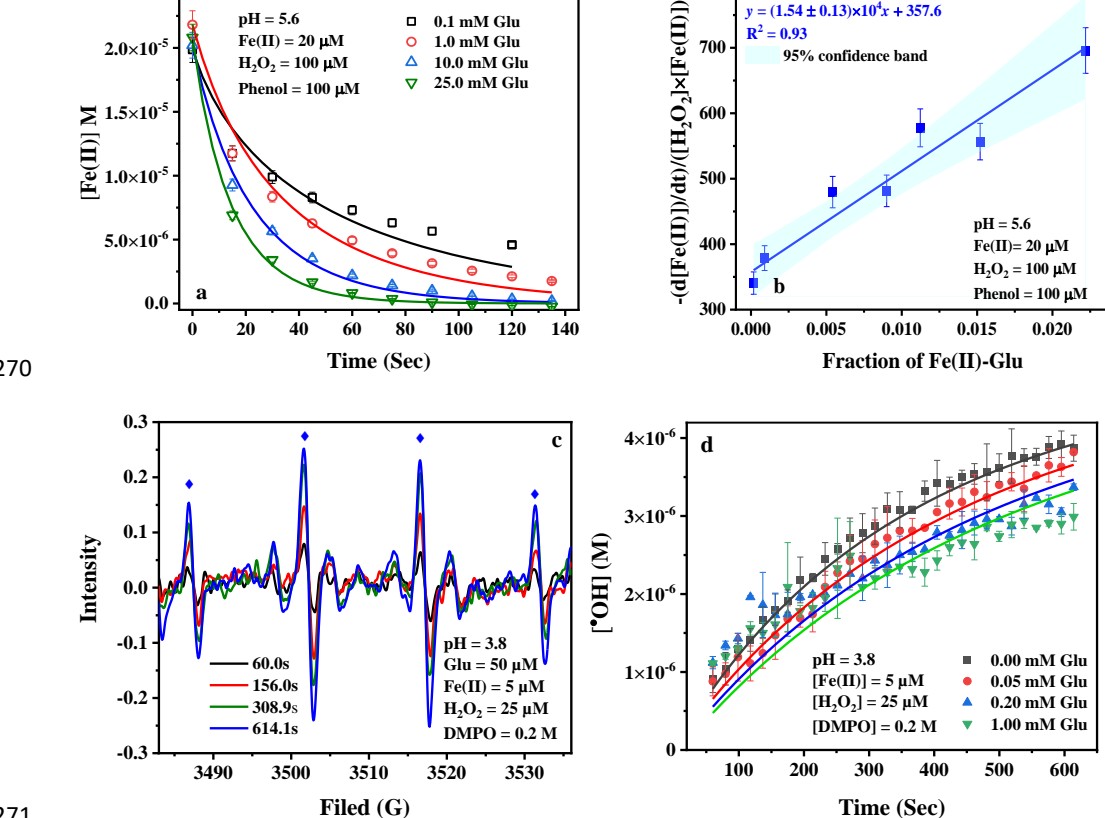


*Fig. 1 Effect of different concentrations of Glu on the kinetics of Fenton reaction (a), apparent*
*rate constant as a function of the fraction of Fe(II)-Glu (b), Signal of EPR corresponding to*
*DMPO-OH (The symbol "◆" marks the position of the characteristic 1:2:2:1 EPR signal of*
*the DMPO-OH adduct.) (c), the kinetics of ●OH generation in Fenton reaction in the presence*



*of different concentrations of Glu, (d). Points are determined experimentally, and lines in*
*figures **a** and **d** are the fit of data using the kinetic model.*
**3.2.2. •OH quantification**
To study the effect of Glu on the •OH generation, EPR experiments were carried out. **Fig. 1c**
shows the EPR signal of DMPO-OH (1:2:2:1) increases with the reaction time, indicating that
•OH is continuously generated. In **Fig. 1d**, the concentration of generated •OH decreases when
the Glu concentration increases from 0 to 1.0 mM. This trend suggests no direct •OH generation
occurs from the reaction of Fe(II)-Glu with $H_2O_2$. This hypothesis has been verified by
employing a kinetic model. The experimental data can be well fitted using the experimental rate
constant $k_{Fe(II)-Glu/H_2O_2} = 1.54 \pm 0.13 \times 10^4$ $M^{-1}$ $s^{-1}$.
**3.3. Fe(III) photolysis in the presence of Glu**
**3.3.1. Fe(II) formation**
To study the effect of Glu on the kinetics and determine the quantum yield of the photolysis of
Fe(III), the photo-driven reaction was carried out in the presence of different concentrations of
Glu (0-200 μM) under simulated solar light. **Fig. SM5a** shows that the Fe(II) generation rate
decreases when the Glu concentration increases, which indicates that Glu slightly reduces the
photoactivity of Fe(III). As shown in **Fig. 2a**, plotting the apparent quantum yield of Fe(II),
$\Phi_{Fe(II)}^{obs}$, as a function of the fraction of Fe(III)-Glu complex, the quantum yield of Fe(II)
decreases with the fraction of Fe(III)-Glu complex increasing. The linear fit can depict the
kinetic data well with a regression coefficient equal to 0.99. As mentioned in **SM2**, the intercept
represents the Fe(II) quantum yield of Fe(III) photolysis under polychromatic irradiation, which



is equal to $0.216 \pm 0.004$. This result is consistent with previous data (Bossmann et al., 1998).
The slope represents the difference between Fe(II) quantum yield of the Fe(III) photolysis and
the value of the photolysis of Fe(III)-Glu complex ($\Phi_{Fe(III)-Glu}^{Fe(II)} - \Phi_{Fe(III)}^{Fe(II)}$), which is equal to -
0.179, hence the Fe(II) quantum yield during the photolysis of Fe(III)-Glu is calculated to be
$0.037 \pm 0.004$. Weller et al.(Weller et al., 2013) investigated the photolysis of Fe(III)-
carboxylate complexes and found the quantum yield of Fe(II) formation from Fe(III)-malonate
at 308 nm and 351 nm, with values of $0.024 \pm 0.001$ and $0.040 \pm 0.003$ respectively. This
suggests that Fe(III) complexes containing unsubstituted carboxylates as a functional group
exhibit lower quantum yields compared to Fe(III).

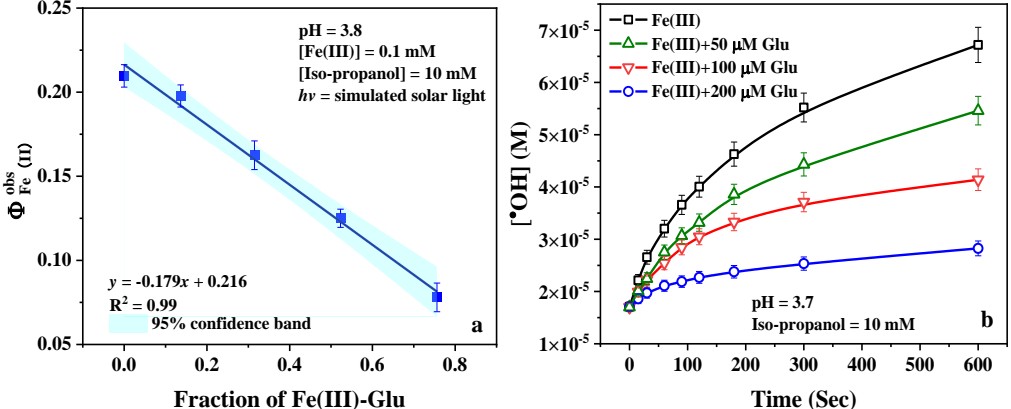


**Fig. 2** *a) The quantum yield of Fe(III) photolysis as a function of the fraction of Fe(III)-Glu*

*complex; b) The •OH generation of Fe(III) photolysis in the presence of different concentrations*

*of Glu. The continuous lines are visual guides generated by applying the "Connect B-Spline"*

*function in Origin 2019.*

**3.3.2. •OH generation**



Since the photolysis of Fe(III) is an important process affecting the budget of $^\bullet$OH in the
atmosphere (Guo et al., 2014), the effect of Glu on the $^\bullet$OH produced by the photolysis process
of Fe(III) was investigated. As shown in **Fig. SM5b,** the acetone generation rate decreases when
the Glu concentration increases, indicating that the $^\bullet$OH generation of the Fe(III) photolysis
decreases in the presence of Glu (**Fig. 2b**). The most likely reason for this observation is the
decrease of the Fe(III) hydroxy complexes (**Table SM5**), hence the decrease of the $^\bullet$OH yield
as the Fe(III)-Glu does not produce $^\bullet$OH directly, but instead forms Glu oxidation products
(Glu$_{ox}$) through the LMCT process. These Glu oxidation products can complex Fe(II) and
regenerate Fe(III), a mechanism known as "the quenching mechanism" proposed by Wang et
al (2010)(Wang et al., 2010). This process reduces the apparent quantum yield of Fe(II) to 0.037
$\pm$ 0.004. This result illustrates that $^\bullet$OH generation could be less in the presence of amino acids
during the daytime in the atmosphere.
**3.4. The Glu fate in the presence of Fe(III) under simulated solar light**
**3.4.1 Photodegradation of Glutamic acid in different systems**
All the above results indicate that Glu not only stabilizes Fe(III)/Fe(II) at higher pH but also
influences the Fenton reaction and photolysis of Fe(III) processes. The main effects were that
the complexes altered the individual reaction rate constants and $^\bullet$OH production.
On the other hand, Glu as the organic ligand can also be degraded during the reaction, especially
photo-reaction in the atmosphere. **Fig. 3** shows the photodegradation kinetics of Glu in different
systems, and the first-order fitted data is reported in **Fig. SM6**. As expected, when only Glu
was present in the solution, no significant degradation was observed after 1 hour of irradiation,
as shown by the UV-Vis spectrum (**Fig. SM2**) of Glu, as there is no significant absorption of





solar radiation. The Glu degradation efficiency slightly increased in the presence of 1 mM $H_2O_2$
with a degradation constant of $2.44 \pm 0.45 \times 10^{-5}$ s$^{-1}$ corresponding with a degradation of 8.5 %
in 1 hour, which is due to the formation of $^{\bullet}OH$ radicals via the photolysis of $H_2O_2$. In addition,
the rate constant of $^{\bullet}OH$ with Glu is $2.3 \times 10^8$ M$^{-1}$ s$^{-1}$ (Masuda et al., 1973), which means that
the reaction between those two components is one of the most important processes for the
degradation of Glu. Considering the second reaction rate constant between $^{\bullet}OH$ and $H_2O_2$
($k_{H_2O_2}^{\bullet OH} = 2.7 \times 10^7$ M$^{-1}$ s$^{-1}$) (Christensen et al., 1982), it can be argued that under adopted
conditions, about 55 % of generated $^{\bullet}OH$ was quenched by the $H_2O_2$, which led to the formation
of less reactive hydroperoxyl radical/superoxide anion pair ($HO_2^{\bullet}/O_2^{\bullet-}$).

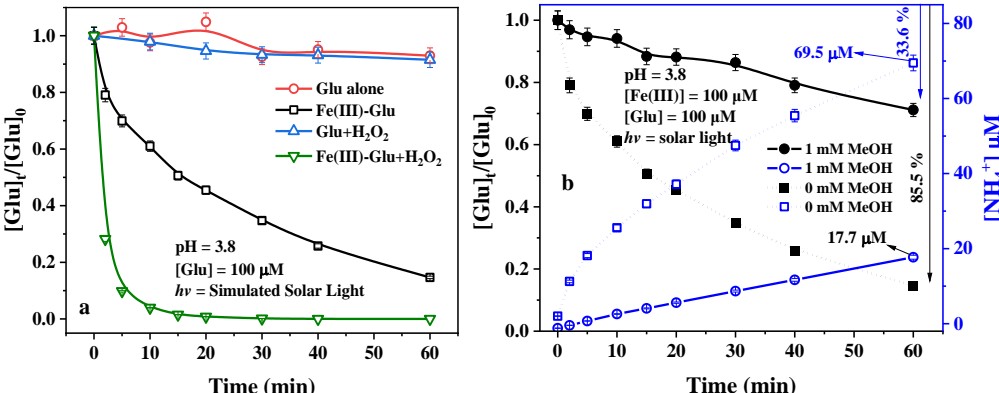


*Fig. 3 Photodegradation of Glu in different systems: Fe(III)-Glu; Glu+H$_2$O$_2$ and Fe(III)-Glu+H$_2$O$_2$([Glu] = 100 µM, [Fe(III)] = 100 µM, [H$_2$O$_2$] = 1 mM). The continuous lines are visual guides generated by applying the "Connect B-Spline" function in Origin 2019.*

Moreover, in the presence of Fe(III), the mixture of Fe(III)-hydroxy and Fe(III)-Glu complexes
underwent the photolysis process. As shown in **Fig. 3a**, about 85 % of Glu was degraded with
a first-order rate constant of $4.99 \pm 0.24 \times 10^{-4}$ s$^{-1}$ after 1 hour of irradiation. This high efficiency



is likely due to two different Glu degradation pathways, one is due to the reaction between Glu
and the $^\bullet$OH radicals generated by photolysis of Fe(III) (R1 and R2), and the other one is due
to the direct photolysis of Fe(III)-Glu leading to the formation of Fe(II) and oxidation products
of the organic ligand (Glu$_{ox}$). The synergistic effect of those two processes highly improved the
Glu degradation efficiency. To distinguish between the contributions of the two degradation
pathways, methanol was selected as $^\bullet$OH scavenger ($k_{\bullet OH}^{Methanol} = 9.7 \times 10^8 \, M^{-1} \, s^{-1}$) (Buxton et al.,
1988). As illustrated in **Fig. 3b**, Glu degradation was inhibited by 60 %, indicating that 40 %
of Glu degradation originates from the photolysis of Fe(III)-Glu complexes. Interestingly, this
ratio aligns with the proportion of Fe(III) and Fe(III)-Glu complexes in the system (**Table SM5**),
confirming the aforementioned conclusion. Furthermore, the degradation of Glu resulting from
the photolysis of Fe(III)-Glu complexes likely does not involve a $^\bullet$OH process (Sun et al., 1998;
Weller et al., 2013).
Glu degradation was observed to be approximately 100 % after 20 mins of irradiation in the
presence of Fe(III) and $H_2O_2$, with a first-order rate constant of $5.13 \pm 1.03 \times 10^{-3} \, s^{-1}$. Compared
to conditions with only Fe(III) or $H_2O_2$, the efficiency of Glu degradation significantly improves
due to the photo-Fenton reaction in the system, which greatly accelerates the formation rate of
reactive species and consequently enhances the degradation rate of Glu.
**3.4.2. Analysis of photodegradation products of glutamic acid**
To distinguish the Glu degradation processes resulting from the photolysis of Fe(III)-Glu
complexes and from those caused by $^\bullet$OH attack, which might lead to the formation of different
products, a series of experiments were conducted. In all cases, IC-MS was employed to analyze
the formation of short-chain carboxylic acid and ammonium ions, providing a deeper





understanding of the photochemical reaction products in various systems under simulated solar
light.
**Figure 4a** depicts the formation of ammonium ($NH_4^+$) in different systems under irradiation. A
positive correlation is observed between the rate of $NH_4^+$ production and the rate of Glu
degradation in various systems, suggesting the occurrence of deamination during the Glu
degradation. Additionally, several carboxylic acids (i.e. acetic, formic, succinic, malonic, and
oxalic acids) were detected (**Table SM6**), as illustrated in **Fig. 4b**, **4c**, and **4d**. Notably, the
concentration of generated carboxyl acids is considerably lower than that of $NH_4^+$.
After 120 min of irradiation, low concentrations of generated $NH_4^+$ and carboxylic acid were
determined during Glu photolysis due to small Glu degradation (see **Fig. 4a** and **Fig. SM7**). In
the presence of 1 mM $H_2O_2$, $NH_4^+$ concentration increased to 7.8 µM within 120 min,
representing a 3-fold increase compared to that produced during Glu photolysis. **Fig. 4b**
demonstrates the formation of carboxylic acids with formate and succinate as primary
carboxylate products, while a negligible concentration of acetate (less than 1 µM) was also
detected, all of which are products of •OH attack.
In the presence of Fe(III), $NH_4^+$ concentration increased to 69.5 µM (**Fig. 4a**) within 60 min.
Simultaneously, the generation of carboxyl acids, such as formate, acetate, and oxalate was
observed. The concentration of formate initially increased, reaching a plateau value of 8.7 µM
at 20 min, after decreasing to approximately 6.4 µM at 60 min. The reason for the decline is
probably the reaction to photo-generated •OH. Acetate concentration increased throughout the
reaction, reaching 10.9 µM at 60 min. Other carboxylates, such as succinate, malonate, and
oxalate, were found in lower concentration, with a maximum of around 2 µM within 5 min. As
mentioned above, in the presence of Fe(III), the Glu degradation can be attributed to two



pathways: one resulting from $^\bullet$OH attack, on the other from the photolysis of the Fe(III)-Glu
complexes.

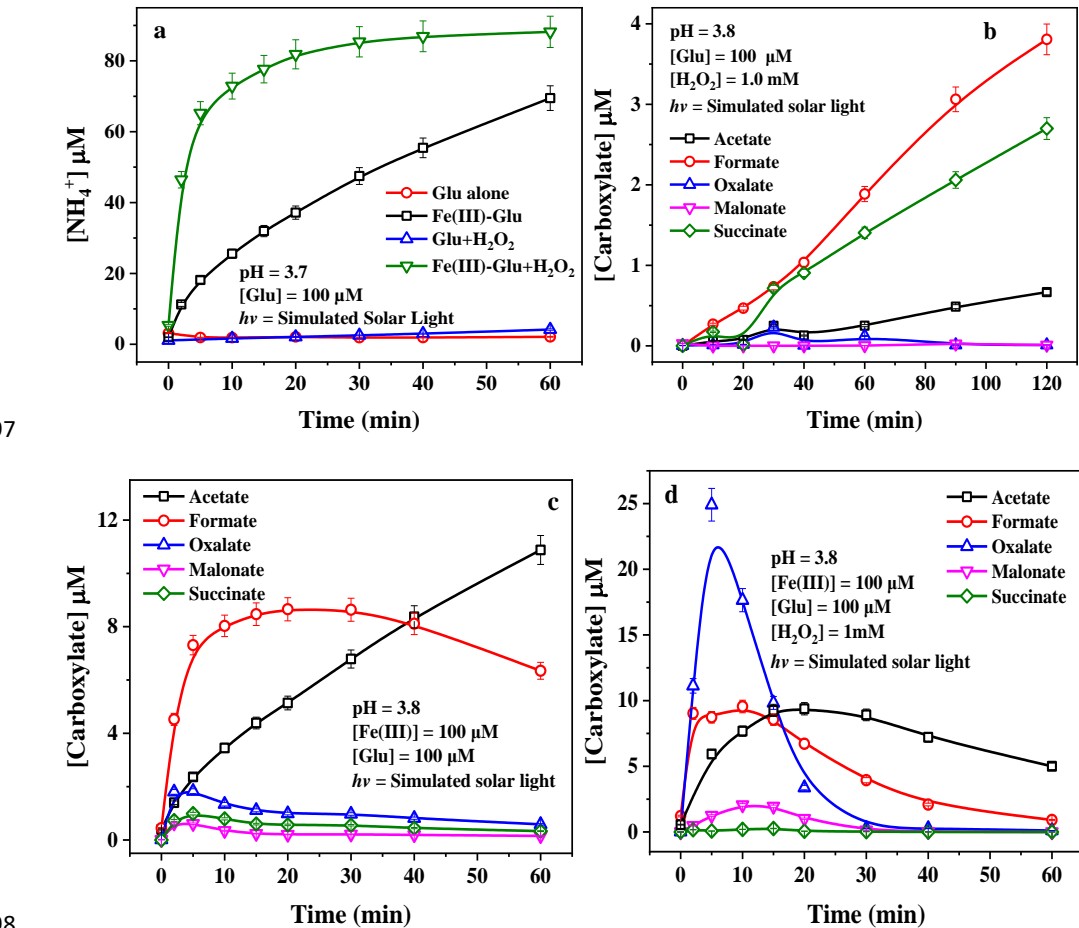



**Fig. 4**. *The by-products of Glu degradation under solar light a) formation of $NH_4^+$ in different*

*systems; formation of carboxylic acids b) in the system Glu + $H_2O_2$; c) in the system Glu +*
*Fe(III) and d) in the system Glu + Fe(III) + $H_2O_2$. The continuous lines are visual guides*
*generated by applying the "Connect B-Spline" function in Origin 2019.*
To distinguish the contribution of these two pathways, isopropanol was employed to quench
$^\bullet$OH in solution generating acetone as the main product (Motohashi and Saito, 1993). As shown



in **Fig. SM8**, only acetate and formate were generated (succinate, malonate, and oxalate were
not detected). Moreover, the presence of isopropanol significantly enhanced the formation of
acetate compared to values observed with only Fe(III) and Glu. This is likely due to the H-
donor effect of the added alcohol or to the reaction between acetic acid radicals (HOOCCH$_2$$^\bullet$)
and HO$_2$$^\bullet$ radicals, the latter being generated through the reaction of $^\bullet$OH with the alcohol. As
shown in **Fig. SM8**, the concentration of generated formate in the presence of isopropanol and
Fe(III) is lower than that when only Fe(III) is added, suggesting that formate was likely not a
primary product generated from the photolysis of Fe(III)-Glu complexes but rather may be
produced by $^\bullet$OH attack of other carboxylic acids. For example, the generated acetate can be
further oxidized reacting with $^\bullet$OH leading to the formation of formate.
This finding is consistent with the result observed in the presence of H$_2$O$_2$ alone (**Fig. SM7**). In
the presence of $^\bullet$OH scavenger, the generation of NH$_4$$^+$ was strongly inhibited with the
formation of 17.7 µM instead of 69.5 µM after 1 h (as previously reported in **Fig. 3b**), which
indicates that the NH$_4$$^+$ formation is mainly due to the $^\bullet$OH attack process. Furthermore, a
significant NH$_4$$^+$ (up to 69.5 µM within 60 min) can be observed in the presence of both Fe(III)
and H$_2$O$_2$ (**Fig.4d**). Oxalate, acetate, and formate were observed as the predominant carboxylate
products with higher concentrations, reaching 24.9, 9.4, and 9.6 µM respectively, before
decreasing. Additionally, the formations of malonate (2.1 µM) and succinate (0.3 µM) were
observed at lower concentrations during the photoreaction. In the presence of H$_2$O$_2$ and Fe(III),
the Fe(III)/Fe(II)-cycle is enhanced via the photo-Fenton reaction. Fe(II) is rapidly re-oxidized
to Fe(III) to produce $^\bullet$OH, which then directly attacks Glu, leading to degradation. Fe(III) is re-
complexed by Glu reactivating the photoreaction and then the iron cycle. Therefore, the
addition of H$_2$O$_2$ favors deamination as well as various carbon-centered radical combination





interactions. The rapid depletion of oxalate after 30 min implies that photolysis of complexes
between Fe(III) and polycarboxylic acid also occurs in this system, while formate, acetate, and
malonate exhibit similar tendencies with different reaction rates. To verify the mineralization
of Glu during the reaction, a TOC was followed during the reaction. As shown in **Fig. SM9a**,
the mineralization efficiency of Glu in the presence of Fe(III) and $H_2O_2$ is significantly higher
than that observed when only Fe(III) is present, due to the presence of the photo-Fenton process.
This finding is consistent with the degradation efficiency of Glu presented in **Figure 3a**. Hence,
these results illustrate that Glu was mineralized to form $CO_2$ and $H_2O$. Moreover, the TOC
values obtained experimentally are higher than the values calculated from the concentration of
Glu and carboxylic acid products, indicating the presence of other organic compounds in the
system. Along these organic substances cannot be detected under our experimental conditions,
they will enter the cloud water gas phase, further participating in atmospheric photochemical
reactions and eventually being mineralized into $H_2O$ and $CO_2$. In the presence of $H_2O_2$, as Glu
undergoes photodegradation, the concentration of $H_2O_2$ in the system continues to decrease
until it is completely consumed (**Fig. SM9b**).
**3.5. Insight into the mechanism of Glu transformation**
The light-driven transformation mechanism of Glu in the presence of Fe(III) was investigated,
with a focus on the $^\bullet OH$-mediated and the ligand-to-metal charge transfer (LMCT) process. The
key difference between the two processes lies in the generation of glutamate radials: the $^\bullet OH$-
mediated process involves a free radical mechanism initiated by hydrogen abstraction, whereas
the LMCT pathway proceeds via an electron transfer process driven by photoexcitation. To
provide a clear comparison, the two mechanisms are illustrated separately in **Scheme. 1**





summarizing the possible Glu degradation pathway, derived from IC-MS analysis of the
detected products.
In the •OH-mediated process, the α- carbon of Glu is identified as the primary site attacked for
•OH attack, initiating the transformation process. Hydrogen abstraction by •OH results in the
formation of glutamate alkyl radical (R-C•(COO⁻)NH₃⁺) and $H_2O$. Subsequently, this alkyl
radical reacts with $O_2$ to generate the alkylperoxy radical (ROO•), which is further converted to
alkoxy radical (RO•) (Goldman et al., 2021; von Sonntag and Schuchmann, 1991). The
formation of RO• is followed by a deamination process, which leads to the formation of
ammonium ($NH_4^+$) and 2-oxoadipic acid through the cleavage of the amino group. Due to the
presence of an oxo group (C=O) adjacent to a carboxyl group (COOH), 2-oxoadipic acid is
chemically unstable and prone to self-decomposition via decarboxylation, resulting in the
formation of succinic acid (Penteado et al., 2019). Further oxidation of succinic acid produces
smaller carboxylic acids.
In contrast, the LMCT process is initiated upon irradiation, resulting in the reduction of Fe(III)
to Fe(II) and the generation of a radical centered on the oxygen atom of α-carboxyl group of
glutamic acid (R-CH(NH₃⁺)C⁺O•O⁻). This high reactive radical undergoes a decarboxylation
process resulting in the formation of an alkyl radical (R-CH•NH₃⁺). Subsequently, the radical
chain reaction propagates in the presence of $O_2$, leading to the formation of smaller carboxylic
acids. It is critical to highlight that the only carboxylic acids detected under the same conditions
are formic acid and acetic acid. This is different from the •OH-mediated process, in which
succinate is first formed and then further decomposed into compounds such as other small
molecular carboxylic acids.





Scheme 1. *The mechanism of Glu degradation in the presence of Fe(III): by* •OH *attacking process and LMCT process. Products in orange are detected by IC-MS.*

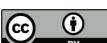



## 4. ATMOSPHERIC IMPLICATION


This study systematically investigated the complexation of Glu with Fe(II)/Fe(III), its effect on
the typical atmospheric reactions (Fenton reaction and Fe(III) photolysis), and its fate in the
atmospheric aqueous phase. Our findings reveal that iron-amino acid complexes (Fe-AAs)
significantly modify the Fe(II)/Fe(III) cycle and $^\bullet$OH budget, diverging from the "classic"
photo-Fenton mechanisms. Specifically, Fe(II)-Glu reacts with $H_2O_2$ at a rate constant two
orders of magnitude higher than Fe(II) alone, potentially improving the iron cycle. Conversely,
Fe(III)-Glu exhibits a lower quantum yield under irradiation, suppressing the Fe(III)/Fe(II)
cycle. Moreover, both reactions result in lower $^\bullet$OH generation, as they favor the formation of
Glu oxidation products ($Glu_{ox}$) over $^\bullet$OH, thus partially affecting atmospheric oxidative
capacity.
To date, the concentration of the Fe(II)/Fe(III)-Glu in cloud water has not yet been directly
measured, hence, based on the reported mean concentrations of Glu (87 nM) (Renard et al.,
2022), Fe(II) (1 µM) (Deguillaume et al., 2014), and Fe(III) (0.5 µM) (Deguillaume et al., 2014)
in cloud water from the Puy de Dôme station (PUY - France), the fraction of Fe(II)-Glu and
Fe(III)-Glu was calculated to be around $8.7 \times 10^{-10}$ - $2.1 \times 10^{-4}$ % and $6.1 \times 10^{-2}$ - $2.4 \times 10^{-1}$ % using
Hyss software at pH = 3 - 7, respectively. Although the fraction of iron-Glu is likely low in
cloud water conditions, the concentration of Glu in cloud droplets may increase during the cloud
water evaporation, leading to an increase in the proportion of iron-Glu complexes. This shift
could alter atmospheric Fenton reaction dynamics, reducing •OH production, particularly at
night (Galloway et al., 2014; Shulman et al., 1997). Similarly, the lower quantum yield of the
Fe(III)-Glu under irradiation inhibits the Fe(II)/Fe(III) cycle and $^\bullet$OH generation especially in
daytime conditions. In addition, recent studies reported that the average AAs contribution





corresponded to 9.1 % of the dissolved organic carbon (DOC) (Bianco et al., 2016), highlighting
their significance. Hence, Fe-AAs play a crucial role in iron speciation, stability, and $^\bullet$OH
budget in atmospheric aqueous phases, which suggests that the inclusion of iron- Fe-AAs in
atmospheric aqueous phase models is essential for a more precise estimation of $^\bullet$OH production,
which is central to understanding oxidation processes and secondary aerosol formation.
In addition, irradiation of Glu in the presence of Fe(III), demonstrated two different
mechanisms ($^\bullet$OH mediated and LMCT process) leading to the generation of different products,
which can further influence the atmospheric chemical composition. Overall, the generation of
$NH_4^+$ is regarded as a link between organic nitrogen species and inorganic nitrogen in cloud
water. The generation of carboxylic acids further increases atmospheric complexity, as the
generated carboxylic acids (e.g., oxalic acid) can be complex with iron and participate in the
consequent photoreactions. In fact, atmospheric models often simplify the distribution and
interactions of transition metal ions (TMIs) with organic compounds, including AAs. This study
highlights the crucial role of the LMCT process in AAs oxidation, which could be considered
in atmospheric modeling.
**ACKNOWLEDGMENT**
This work was supported by the Agence Nationale de la Recherche of France in the frame of
the PRCI project REACTE.
**AUTHOR CONTRIBUTION**
**Peng Cheng:** Investigation, Formal analysis, Writing – original draft**; Gilles Mailhot:** Funding
acquisition, Review & editing, Supervision**; Mohamed Sarakha:** Review & editing,
Supervision**, Guillaume Voyard:** Technical support**; Daniele Scheres Firak:** Review &
editing**; Thomas Schaefer:** Funding acquisition, review & editing**; Hartmut Herrmann:**



Review & editing; **Marcello Brigante:** Conceptualization, Writing-review & editing,
Supervision.

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
