# Peer review of "The Effect of Amino Acids on the Fenton and"

_EGUsphere, 2025_

## Author Comment (AC1)

**Reviewer 1**

This is a well-written and comprehensive study investigating the complexation of glutamic acid (Glu) with Fe(II)/Fe(III) and its implications for atmospheric chemistry, particularly in cloud water. The experiments are well-designed, and the results provide valuable insights into the role of amino acids in modifying Fenton and photo-Fenton reactions. The findings are novel and contribute significantly to our understanding of atmospheric oxidative processes.

- The atmospheric relevance could be further emphasized by discussing how variations in cloud pH, light intensity, or iron/ligand ratios might influence the observed processes.

  **Response**

  We appreciate the reviewer's insightful suggestion to strengthen the discussion of the atmospheric relevance of our findings. In the revised manuscript, we have added a paragraph discussing how typical variations in cloud water pH (usually ranging from 3 to 7), diurnal changes in solar radiation, and fluctuating iron-to-ligand ratios can modulate the reaction kinetics and pathways of the studied system. This addition highlights the broader implications of our results for cloud and aerosol chemistry under realistic atmospheric conditions. The new discussion can be found on **page 26-27, lines 510-518 and lines 522-526**.

- It would be better to explicitly state in the first paragraph of "Introduction' that the role of amino acids in modifying iron redox chemistry and OH production remains poorly understood.

  **Response**

  We have revised the introduction, this sentence has been added in the first paragraph. The modification can be found at **lines 50-54**.

- Were the pH conditions (3.8–5.6) chosen to represent specific atmospheric scenarios (e.g., polluted vs. remote clouds)? A brief justification would be useful.

  **Response**

  The pH values of 3.8 and 5.6 were chosen based on both experimental needs and atmospheric relevance. pH 5.6 was used in kinetic studies to promote Fe(II)-Glu complex formation, while pH 3.8 was used in ESR experiments where •OH detection is more effective under acidic conditions. These values fall within the typical pH range of cloud water and represent different environments—pH 3.8 for polluted regions and pH 5.6 for remote areas. This allows us to study the system under realistic atmospheric conditions. A clarification has been added in the revised manuscript at **lines 133–139**.

- For the photolysis experiments, was the light spectrum adjusted to match real solar conditions?

  **Response**

  The light spectrum was not specifically adjusted, but the spectral distribution is similar to natural solar radiation, which is important for the photoreactions of Fe(III) and Fe(III)-Glu, As shown in Fig. SM2 and our previous publication. The modification is provided at **lines 169-173**.

- The reported rate constant ($1.54 \times 10^4$ M-1 s-1) is a key finding. However, how does this compare with other Fe(II)-organic complexes (e.g., oxalate, citrate)? A brief discussion would be useful.

  **Response**

  We did the comparison with our recently published value of Fe(II)-Oxalate, a brief discussion was added as well. The modifications were provided at **lines 271-276.**

- The detection of formate/acetate as primary LMCT products is interesting. Could these compounds further complex Fe(III) and influence subsequent reactions?

  **Response**

Yes, you are right. As we wrote in the part of atmospheric implications at **lines 537-539**, those generated compounds can further complex with iron and participate in the consequent photoreactions.

- The discussion of Fe-Glu fractions in cloud water is insightful, but how might these vary in highly polluted vs. marine environments?

**Response**

The Fe-Glu fraction in cloud water is expected to vary depending on the environmental context. In highly polluted environments, elevated levels of iron and amino acid (such as Glu from biomass burning or urban sources) may promote the formation of Fe-Glu complexes. In contrast, marine clouds often contain lower concentrations of both Fe and amino acid (like Glu is around 33 pM collected in Venice on the Sacca San Biagio Island), leading to a smaller Fe-Glu fraction. The more discussion was provided in atmospheric implications part at **lines 510-518**.

- How might these findings affect our understanding of SOA formation?

**Response**

Thank you for the insightful question. Our findings suggest that Fe(III)-Glu complexes can influence aqueous-phase SOA formation in two key ways. First, the photooxidation of glutamic acid generates small organic acids (e.g., formate, acetate) and ammonium, which are known precursors of SOA in cloud water. Second, the presence of Fe-Glu alters the generation of •OH radicals and introduces additional LMCT-driven pathways, potentially modifying the oxidation chemistry of water-soluble organic compounds. These results highlight a previously underappreciated mechanism by which metal–amino acid complexes may affect the oxidative capacity of cloud water and contribute to SOA formation, particularly under polluted or organic-rich atmospheric conditions. The more discussion was provided in atmospheric implications part at **lines 528-535**.

---

## Author Comment (AC2)

**Reviewer 2**

The article presents a detailed investigation into the potential effect of glutamic acid (Glu)-iron complexes on Fenton and photo-Fenton reactions in cloud water and on the hydroxyl radical yield. Using Glu as a model amino acid, the study offers compelling insights into how these organic compounds may interfere with iron (photo)chemistry leading to an impact on the oxidative potential of cloud water. Overall, this study makes a valuable contribution to our understanding of aqueous-phase atmospheric chemistry. I have just few comments.

- It would be interesting to compare the results with other well-known iron complexes with organic ligands that might be relevant for cloud water (for example Fe-oxalate)

   **Response**

   We fully agree that comparing our results with well-known iron-organic complexes relevant to cloud water, such as Fe-oxalate, is important. Accordingly, we have revised the manuscript to include a discussion on this aspect to enhance the environmental relevance of our study. The modification was given at **lines 271-276**.

- It would be good to further discuss the potential of forming the Glu-iron complexes in real cloud water. Are there other compounds that may complex strongly the iron? Since the Glu is only a model for AAs, I think the work is relevant because other AA-iron complexes may behaviour in the same way.

   **Response**

   The new discussion of iron-Glu complexes in real cloud water was provided and reported in **lines 510-518**. Indeed, other compounds commonly present in cloud water, such as various amino acids and low-molecular-weight carboxylic acids, are known to form stable complexes with iron and may undergo similar (photo)-Fenton reactions as observed with glutamic acid. The corresponding modifications can be found in the revised manuscript at **lines 532-535.**

- In section 2.2.1 please add the concentration of H2O2 used in the experiments (line 131).

   **Response**

   The $H_2O_2$ concentration was added in the section 2.2.1, shown at **lines 140-142**.

- In Figure 1 the letters (a, b, c and d) referred to the different plots are not very visible, probably it would be good to put them outside the plots.

   **Response**

   To maintain a consistent figure style throughout the manuscript, we did not move the panel labels (a, b, c, d) outside the plots. However, we increased their font size to enhance visibility and ensure they are clearly distinguishable. This modification can be found at **line 285** in the revised manuscript.

- From Figure 3 and Figure MS6 the degradation of Glu with and without H2O2 seems very similar, please report also the data about the irradiation of Glu alone.

   **Response**

   The similar degradation of Glu with and without 1 mM $H_2O_2$ is due to the low light absorption of $H_2O_2$ and limited •OH generation. We also tested Glu under irradiation alone, and the rate constant was $k = (1.90 \pm 0.22) \times 10^{-5}$ s$^{-1}$, indicating negligible degradation. This is expected, as Glu does not absorb solar light. The data have been added to the revised manuscript (**lines 347-350**)

- In the figure caption for Figure 3 is described only the left panel and not the right panel, please add it.

   **Response**

   Thank you for very careful review. The description for the right panel is added as shown in **lines 360-363.**

- Page 19, lines 389-391, please add the reactivity constant between formic acid and hydroxyl radical.

**Response**

The reactivity constant between formic and OH radical was added and shown **on page 20 at line 409**.

- I suggest to change the colours of the lines in the figure 4a, to avoid confusion with the other part of the figures (Figure4b,c,d) where for each colour indicates a specific carboxylic acid.

**Response**

Thank you for your suggestion. We have changed the colors of the lines in the figures 4a. The modification can be found at **line 414**.

- In page 22 you discuss the TOC results, have you also performed IC (inorganic carbon) analysis? From that you could also directly measure the mineralization of the Glu. If you have these data, please add them in the manuscript (or in the SI).

**Response**

We provided measurement of total carbon (TC) and inorganic carbon (IC) and calculated the total organic carbon (TOC) accordingly. However, since the reaction system is open and acidic, a significant portion of the $CO_2$ produced during glutamate mineralization escapes from the solution, leading to an underestimation of the actual mineralization degree when using IC data. For this reason, we did not include the IC results in the manuscript. We will clarify this point in the revised SI (SM5) to avoid misunderstanding.

- Page 23, lines 456-457, please add a reference for the deamination process as you did for the other reaction involved in the overall mechanism.

**Response**

A new reference was added, the modification can be found at **lines 476-477.**

---

## Author Response (AR2)

**l. 169: Which "Energy" is meant here? The unit μW cm-2 seems to indicate that this is a power density (not energy), likely referring to radiant flux density / irradiance. Please specify.**

**Response**

Thank you for your suggestion. You are absolutely right — we have revised the manuscript by replacing "energy" with "irradiance", which is more accurate and appropriate in this context. The modification can be found at lines 169-170.

**l. 271: If you use the expression "significantly higher", please indicate the level of statistical significance. Otherwise, consider using "much higher".**

**Response**

Thank you for the reminder. We have revised the expression "significantly higher" to "much higher" in the manuscript. (Shown at line 271)

**l. 273: What is referred to with "This results is very close to our recently published data as well"? It is not clear whether "this result" refers to the determined rate of Fe-Glu, or published rate coefficients of the Fenton reaction. Neither of them seems "very close".**

**Response**

Thank you for the helpful suggestion. We have clarified this part to avoid ambiguity. The revised text now reads:

Moreover, the value is about five times higher than our recently reported value for the reaction between Fe(II)-oxalate and $H_2O_2$ ($3.2 \pm 0.3 \times 10^3$ $M^{-1}$ $s^{-1}$) (Scheres Firak et al., 2025). Despite the quantitative difference, both two obtained values are of the same order, highlighting the significant reactivity enhancement conferred by organic ligand coordination. (lines 273-277)

**l. 274: "reactivity constant" should probably read "rate constant" to align with the rest of the manuscript, unless a different concept is used here.a**

**Response**

Thank you for pointing this out. We have corrected the term "reactivity constant" to "rate constant" throughout the manuscript (lines 150 and 220, 274,), to maintain consistency and clarity.

**l. 276: "hence" should probably read "enhance"**

**Response**

Thank you for your suggestions. We have revised this paragraph. The modification can be found at lines 275-277.

**l. 350: Since solar radiation was not used in this experiment, I would suggest to phrase this as "no significant light absorption in the solar spectrum".**

**Response**

Thank you for your suggestions. We have corrected this part by using no significant light absorption in the solar spectrum in the manuscript. (lines 349-353)

**l. 350: How was an "efficiency" determined? Efficiency typically refers to some form of ratio of input and output.**

**Response**

Thank you for your valuable comment. We have removed the word "efficiency" from the sentence, as no quantitative measure of efficiency was provided, making the expression more accurate. (line 350)

**l. 363: "Continuous lines" - Does this expression also include the dashed lines? What is the difference between dashed and solid lines?**

**Response**

Thank you for pointing this out. To avoid confusion, we have modified the figure to use only solid lines and clarified in the caption that the continuous lines are for visual guidance. (line 359).

**l. 409: What is the unit of the reaction rate coefficient provided here? It is not clear why this is given here as the previous sentence refers to production of OH, not consumption of OH by formic acid. A full sentence of explanation might be needed.**

**Response**

Thank you for your suggestion. We have clarified the sentence and added units ($M^{-1}\ s^{-1}$) for the rate constant. The revised sentence now reads: (lines 407-411)

The concentration of formate initially increased, reaching a maximum value of 8.7 μM at 20 min, followed by a decline to about 6.4 μM at 60 min. The reason for the decline is probably due to the reaction of formate with photo-generated $^{\bullet}OH$ ($k_{Formate}^{^{\bullet}OH}$ = 1.3-1.4×$10^8$ $M^{-1}\ s^{-1}$) (Buxton et al., 1988). Acetate concentration steadily increased throughout the reaction, reaching 10.9 μM at 60 min.

**l. 479: "Further oxidation of succinic acid produces smaller carboxylic acids." - Please provide a reference.**

**Response**

Thank you for the suggestion. We have added a relevant reference to support the statement regarding further oxidation of succinic acid. (line 480)

**l. 518: It is not clear what is meant here with "proportion". If you increase the concentration of dissolved material through evaporation of water, would not all organic ligands increase in concentration, and thus the "proportion" of Fe-Glu remain constant?**

**Response**

Thank you for your insightful comment. We have revised the sentence to clarify that while the total concentration of ligands increases due to evaporation, their relative proportions remain constant. (line 518)

**l. 524: Please explain what is meant with "the efficiency and pathways of the observed processes are highly dynamic".**

**Response**

Thank you for the constructive comment. We have clarified this point in the revised manuscript. Specifically, we now explain that the photochemical processes involving Fe–Glu complexes are highly dependent on environmental conditions (e.g., pH, light intensity, ligand concentrations), which in turn affect the degradation and transformation pathways of amino acids. The revised sentence can be found at lines 524–527.

**l. 535: "Overall, the generation of $NH_4^+$ is regarded as a link between organic nitrogen species and inorganic nitrogen in cloud water." - It is not clear if this is a conclusion of this work or a literature reference. Please explain how this is derived from this work or provide a reference.**

**Response**

Thank you for pointing this out. In our work, $NH_4^+$ was detected as one of the degradation products of Glu under irradiation, which suggests a transformation of organic nitrogen into inorganic nitrogen. We have revised the sentence accordingly to clarify that this is an experimental observation from our study and to highlight its atmospheric implication. A reference has also been added to support this link (see lines 536–543).

**l. 538: "can be complex with iron and participate in the consequent photoreactions" should probably read, e.g., "can complex iron and influence photochemistry"**

**Response**

Thank you for the suggestion. We agree and have revised the sentence to: *"can complex iron and influence* consequent *photochemistry"* which improves clarity and aligns with the intended meaning (see lines 541-543).